# Transcriptome Analysis Reveals the Defense Mechanism of Cotton against *Verticillium dahliae* Induced by Hypovirulent Fungus *Gibellulopsis nigrescens* CEF08111

**DOI:** 10.3390/ijms24021480

**Published:** 2023-01-12

**Authors:** Zili Feng, Feng Wei, Hongjie Feng, Yalin Zhang, Lihong Zhao, Jinglong Zhou, Jiatao Xie, Daohong Jiang, Heqin Zhu

**Affiliations:** 1State Key Laboratory of Cotton Biology, Institute of Cotton Research of Chinese Academy of Agricultural Sciences, Anyang 455000, China; 2College of Plant Science and Technology, Huazhong Agricultural University, Wuhan 430070, China; 3Western Agricultural Research Center, Chinese Academy of Agricultural Sciences, Changji 831100, China

**Keywords:** cotton, Verticillium wili, biocontrol, transcriptome, *Gibellulopsis nigrescens*, induced systemic resistance

## Abstract

Verticillium wilt is a kind of plant vascular disease caused by the soilborne fungus *Verticillium dahliae*, which severely limits cotton production. Our previous studies showed that the endophytic fungus *Gibellulopsis nigrescens* CEF08111 can effectively control Verticillium wilt and induce a defense response in cotton plants. However, the comprehensive molecular mechanism governing this response is not yet clear. To study the signaling mechanism induced by strain CEF08111, the transcriptome of cotton seedlings pretreated with CEF08111 was sequenced. The results revealed 249, 3559 and 33 differentially expressed genes (DEGs) at 3, 12 and 48 h post inoculation with CEF08111, respectively. At 12 h post inoculation with CEF08111, Kyoto Encyclopedia of Genes and Genomes (KEGG) enrichment analysis indicated that the DEGs were enriched mainly in the plant–pathogen interaction, mitogen-activated protein kinase (MAPK) signaling pathway-plant, and plant hormone signal transduction pathways. Gene ontology (GO) analysis revealed that these DEGs were enriched mainly in the following terms: response to external stimulus, systemic acquired resistance, kinase activity, phosphotransferase activity, xyloglucan: xyloglucosyl transferase activity, xyloglucan metabolic process, cell wall polysaccharide metabolic process and hemicellulose metabolic process. Moreover, many genes, such as calcium-dependent protein kinase (*CDPK*), flagellin-sensing 2 (*FLS2*), resistance to *Pseudomonas syringae pv. maculicola* 1(*RPM1*) and myelocytomatosis protein 2 (*MYC2*), that regulate crucial points in defense-related pathways were identified and may contribute to *V. dahliae* resistance in cotton. Seven DEGs of the pathway phenylpropanoid biosynthesis were identified by weighted gene co-expression network analysis (WGCNA), and these genes are related to lignin synthesis. The above genes were compared and analyzed, a total of 710 candidate genes that may be related to the resistance of cotton to Verticillium wilt were identified. These results provide a basis for understanding the molecular mechanism by which the biocontrol fungus CEF08111 increases the resistance of cotton to Verticillium wilt.

## 1. Introduction

Cotton Verticillium wilt, caused by the soil-borne fungus *Verticillium dahliae*, poses a major threat to a broad host range of more than 400 plant species [1,2]. The disease is difficult to control because of its long-term survival as microsclerotia in the soil [3]. Breeding resistant varieties is the main method to control cotton Verticillium wilt, but there is no successful cultivar with high resistance because of the lack of effective resistance resource materials [4,5]. To date, no fungicide has been identified to have a viable effect on control of Verticillium wilt of cotton in a field environment [6,7,8].

At present, the use of biological control agents is a promising, more environmentally friendly strategy to control Verticillium wilt of cotton [9]. Numerous microorganisms have been proved to have biocontrol effects on Verticillium wilt. The nonvolatile substances produced by CEF-818 (*Penicillium simplicissimum*), CEF-325 (*Fusarium solani*), CEF-714 (*Leptosphaeria sp.*) and CEF-642 (*Talaromyces flavus*) inhibit *V. dahliae* growth [10]. Cotton endophytic fungus *Chaetomium globosum* CEF-082, which was isolated from upland cotton plants, suppressed the growth of *V. dahliae* and increased cotton resistance to Verticillium wilt [11]. These microorganisms protect plants from the deleterious effects of various pathogens, cause induced systemic resistance (ISR), compete for nutrients and colonization space, or promote plant growth through the production of phytohormones and the delivery of nutrients [12]. Induced plant resistance is one of the means of biological control. ISR refers to the interaction between plant roots and rhizosphere microorganisms, which leads to the formation of plant defense systems and many new chemicals to fight against disease pathogens, such as bacteria, fungi and viruses [13]. Induced resistance is an important method for plants to protect themselves. Numerous studies have shown that various biological control agents can suppress Verticillium wilt in different host species [14]. Iturins mediate the defense response, and significantly activate *PR1*, *LOX* and *PR10* at 24 h after *V. dahliae* infection [15]. *Fusarium oxysporum* 47 (Fo47) reduced the symptoms of Verticillium wilt in pepper, with the expression of three defense genes, *CABPR1*, *CACHI2* and *CASC1*, upregulated in the roots [16]. Bacillus subtilis DZSY21 reduced the disease severity of southern corn leaf blight and upregulated the expression level of *PDF1.2* [17]. Hypovirulent *Verticillium* strains have been previously isolated from cotton and demonstrated to be promising agents for biocontrol of cotton Verticillium wilt [18,19]. Zhao et al. isolated a hypovirulent strain of *Gibellulopsis nigrescens* (strain Vn-1) causing minimal wilt in sunflower, which could be used to control cotton Verticillium wilt [20]. *G. nigrescens*, formerly known as *V. nigrescens*, can cause extremely weak Verticillium wilt, and has been isolated from a variety of crops [21].

Plant innate immunity or basal defense responses are triggered by microbial conserved MAMPs effectors [22,23], or by endogenous DAMPs effectors released from injured host plants [24]. Pattern recognision receptors (PRRs), for example, recognition of receptor-like kinases (RLK) or receptor-like proteins (RLP) with MAMPs/DAMPs can trigger a series of cellular and physiological reactions, including Ca2^+^ surge, extracellular basification, membrane potential depolarization, ion efflux, nitric oxide (NO) release, reactive oxygen species burst, phosphatidic acid accumulation, MAPK cascade activation, ethylene synthesis, callose deposition, immune responses such as the transcription of defense genes, thereby making host plants resistant to a variety of pathogens [25,26,27,28,29,30].

In previous studies, we found that the endophytic fungus *G. nigrescens* CEF08111 isolated from healthy upland cotton plants can significantly improve the resistance of cotton to Verticillium wilt, but has no antagonism against *V. dahliae* [18]. However, the signaling mechanism induced by CEF08111 is unknown. Therefore, the purpose of this study was to reveal the molecular mechanism by which CEF08111 increased cotton resistance to Verticillium wilt via RNA sequence analysis.

## 2. Results

### 2.1. Control Effect of CEF08111 on Verticillium Wilt of Cotton

The disease index was 7.9 in the treatment group (CEF08111+ *V. dahliae*) and 47.9 in the control group (water+ *V. dahliae*) 20 d after *V. dahliae* inoculation (Figure 1a). The results showed that CEF08111 reduced vascular bundle discoloration and enhanced the resistance of cotton to Verticillium wilt (Figure 1b,c).

### 2.2. Content of H_2_O_2_, JA and SA

After the treatment of cotton seedlings with CEF08111 and Vd076, the change of H_2_O_2_ content in cotton leaves was significantly different. The H_2_O_2_ content in CEF08111 treatment group increased slightly within 6 h, and then decreased to the level of sterilized water treatment. The H_2_O_2_ content in Vd076 treatment group continued to increase, reaching the highest level at 48 h (97.11 μmol/g), and then gradually decreased, which was comparable to that in sterilized water treatment at 96 h post inoculation (hpi) (Figure 1d).

The content of JA cotton seedlings increased sharply after cotton seedlings treatment with CEF08111, Vd076 and sterilized water. Subsequently, the JA content of CEF08111 treatment reached the highest at 12 hpi, and then decreased gradually. After vd076 treatment, JA content first increased, then decreased, and then increased (Figure 1e).

The SA content in the CEF08111 treatment group was higher than that in the Vd076 and the sterilized water group throughout the majority of the duration of the experiment and lower than that in the Vd076 group at 96 hpi. The SA content in the CEF08111 group was highest at 6 hpi (1.11 μmol/g) (Figure 1f).

### 2.3. RNA Sequencing and Transcript Identification

To obtain transcriptome profiles of susceptible cotton variety Jimian 11 inoculation by *G. nigrescens* CEF08111 (Gn), *V. dahliae* Vd076 (Vd) and sterile water (SW), respectively, we performed RNA-Seq analysis at 3, 12 and 48 hpi, with three biological replicates performed at each time point for each treatment. In this study, an average of ~6.48 Gb of clean data were generated for each sample using the BGISEQ-500 platform (Appendix A). The minimum correlation between the three replicates was 74.4% (Appendix A). Principal component analysis (PCA) of 27 arrays (Appendix A) was also used to compare the samples and to explore the dynamic changes in the cotton transcriptome after treatment with Gn and Vd. The average clean reads of the 27 samples were 42.27 M. The lowest Q20 value of the clean reads was 95.66, and the lowest Q30 value was 89.50 (Appendix A). A total of 36,551 new transcripts were found, of which 7644 belonged to new protein-coding genes (Appendix A). These data showed that the RNA-Seq quality was applicable for further analysis.

There were 5553 transcription factors (TF) annotated (Appendix A), belonging to 59 families. The largest number of TF are MYB and AP2-EREBP families, with 722 and 540 genes, respectively. A total of 4824 Plant Resistance Genes (PRG) were annotated (Appendix A), including 1395 RLP (receptor-like proteins consists of a leucine-rich receptor-like repeat, a transmembrane region of ~25 AA, and a short cytoplasmic region, with no kinase domain), 790 NL (Contains NBS domain at N-terminal and LRR st the C-terminal, and lack of the CC domain), 643 TNL (contains a central nucleotide-binding subdomain).

### 2.4. DEGs of Cotton Resistance to Verticillium Wilt Induced by CEF08111

DEGs of cotton resistance to Verticillium wilt induced by CEF08111 at 3, 12 and 48 hpi were identified based on an adjusted *p*-value of ≤0.01 and a log2 fold change of ≥2. FPKM (fragments per kilobase of exon per million fragments mapped) values for all genes and the fold changes and adjusted *p*-values for DEGs are shown in Appendix A, respectively.

To delineate the mechanisms of *G. nigrescens* CEF08111 biocontrol of Verticillium wilt in cotton, one comparison group was set, using *V. dahliae* Vd076-treated samples alone, compared with mock treatment samples at different time points. It was observed that whether *G. nigrescens* CEF08111 was treated or *V. dahliae* Vd076 were treated, a large number of genes expression levels in cotton could be changed (Figure 2a). The comparison of transcriptomes was performed for each group. Total number of DEGs (up and down) and their distribution is shown in Figure 2a. Comparative analysis among 3 hpi to 48 hpi showed the highest number of total DEGs at 12 h, induced by CEF08111. There were 5335 upregulated and 6375 downregulated DEGs with sterilized water as control. However, the total number of DEGs induced by Vd076 gradually increased with time. For instance, there were 2383 upregulated and 1724 downregulated DEGs with sterilized water as control at 48 h induced by Vd076.

For the DEGs data of multiple comparison groups, we use a Venn diagram to show the situation of genes among different comparison groups (Figure 2b–d). The results indicated that there are 249 DEGs, 3559 DEGs and 33 DEGs at 3 hpi, 12 hpi and 48 hpi, respectively.

### 2.5. GO Enrichment Analyses of DEGs

To determine the functions of DEGs involved in the response to *G. nigrescens* CEF08111 and *V. dahliae*, we performed GO (Gene Ontology) enrichment analyses using the Phyper function in R software (version 4.2.1). The results showed that DEGs at 3 hpi, the most highly enriched GO terms, were those associated with response to stimulus, including response to external stimulus, systemic acquired resistance, tropism and defense response, incompatible interaction (Figure 3a). Then, for the 3559 DEGs at 12 hpi, significant GO terms were primarily enriched in protein kinase activity, phosphotransferase activity, alcohol group as acceptor, kinase activity and protein serine/threonine kinase activity (Figure 3c). At 48 hpi, the most highly enriched GO terms were those associated with the organization of the cell wall or the metabolism of its components, including xyloglucan: xyloglucosyl transferase activity, xyloglucan metabolic process, cell wall polysaccharide metabolic process and hemicellulose metabolic process (Figure 3e). As the first barrier to invasion, the cell wall is the first obstacle for most pathogens [31]. Therefore, DEGs associated with these significant terms may play important roles against *V. dahliae* infection in cotton and induce cotton to develop resistance to *V. dahliae*.

### 2.6. KEGG Enrichment Analyses of DEGs

We performed KEGG (Kyoto Encyclopedia of Gene and Genomes) functional enrichment analyses using the Phyper function in R software (version 4.2.1). The results showed that DEGs at 3 hpi were mainly significantly enriched in glycometabolism and phenylpropanoid biosynthesis pathways, such as glycolysis/gluconeogenesis, phenylpropanoid biosynthesis, cutin, suberine and wax biosynthesis, amino sugar and nucleotide sugar metabolism, and other glycan degradation and cyanoamino acid metabolism (Q-value < 0.01) (Figure 3b and Table 1).

However, 3559 DEGs at 12 hpi were significantly enriched in pathways related to disease resistance. As shown in Figure 3d, for the plant–pathogen interaction, MAPK signaling pathway—plant and plant hormone signal transduction (Figure 3d and Table 1), there were 39 *FLS2* genes, 17 upregulated and 22 downregulated; 2 *CNGC* genes, 1 upregulated and 1 downregulated; 29 *CaMCML* genes, 1 upregulated and 28 downregulated; 19 calcium-dependent protein kinase (*CDPK*) genes, 1 upregulated and 18 downregulated; and 5 *Rboh* genes, 1 upregulated and 4 downregulated (Figure 4). These genes were related to the metabolism of reactive oxygen species (ROS), Ca^2+^ and NO. In the MAPK signaling pathway—plant pathway, 166 DEGs regulated 31 crucial points related to *FLS2*, H_2_O_2_, ethylene (ET), jasmonic acid (JA), abscisic acid (ABA), ROS and Ca^2+^ (Appendix A). In the plant hormone signal transduction pathway, 154 DEGs were significantly upregulated or downregulated in JA, ET, ABA, brassinosteroid, auxin and gibberellin pathways, which may play an important role in resistance of cotton to *V. dahliae* (Appendix A).

### 2.7. Putative R Genes Involved in Resistance to Verticillium Wilt

On the basis of the transcriptome analysis, a total of 710 candidate genes that may be related to the resistance of cotton to Verticillium wilt were identified (Appendix A), including 210 RLPs (receptor-like proteins consists of a leucine-rich receptor-like repeat, with no kinase domain), 115 NLs (contain NBS domain at N-terminal and LRR at the C-terminal, and lack the CC domain), 2 RPW8-NL (contain NBS, LRR and RPW8 domains), 113 CNLs (contain a central nucleotide-binding subdomain as part of a larger entity called the NB-ARC domain), 91 Ns (contain NBS domain only, lack of LRR), 89 TNLs (contain a central nucleotide-binding subdomain as part of a larger entity called the NB-ARC domain), 49 Ts (contain TIR domain only, lack of LRR or NBS), 14 RLK-GNK2 (RLK class with additional domain GNK2), 10 CNs (contain a central nucleotide-binding subdomain as part of a larger entity called the NB-ARC domain), 7 Mlo-like (a member of the Mlo-like resistant proteins), 5 RLK (consist of an extracellullar leucine-rich repeat region and an intracellular kinase domain), and five other types (consists in a miscellaneous set of R proteins that do not fit into any of the known four classes, but that has a resistance function).

### 2.8. Gene Co-Expression Network Analysis

Weighted gene co-expression network analysis (WGCNA) is a common algorithm used in transcriptomic studies [32]. Five different modules were obtained using a gene dendrogram colored according to correlations between gene expression levels (Figure 5a). Among them, the genes in red, black and magenta were highly expressed in cotton inoculated by CEF08111 at 12 hpi and 48 hpi, respectively (Figure 5b). We performed KEGG analysis for these three modules. For the red module, pathways related to alpha-linolenic acid metabolism was enriched (Appendix A); for black, pathways related to glycolysis/gluconeogenesis, amino sugar and nucleotide sugar metabolism, cutin, suberine and wax biosynthesis, phenylpropanoid biosynthesis and RNA polymerase were enriched (Appendix A); for magenta, pathways related to cutin, suberine and wax biosynthesis, and ubiquinone and other terpenoid-quinone biosynthesis were enriched (Appendix A). Notably, 7 DEGs of the pathway phenylpropanoid biosynthesis in the red, black or magenta module were also present in the EDGs at 3 hpi (Appendix A), and these genes are related to lignin synthesis. Therefore, the disease-resistance genes should be studied in greater depth in the future to elucidate their role in the CEF08111-induced resistance response to *V. dahliae* infection in cotton.

### 2.9. Verification of RNA-Seq Analysis by qRT-PCR

To verify the RNA-Seq data, 12 DEGs were chosen for qRT-PCR; three biological replicates were performed. These 12 genes were selected from significantly enriched KEGG pathways. The expression data obtained by qRT-PCR were consistent with the RNA-Seq results (Appendix A), indicating a similar trend between the transcriptome and qRT-PCR datasets (Figure 6). Among the 12 DEGs, a significantly upregulated gene with ID Ghir_D05G019060.1 (Figure 6f) was predicted to encode a xyloglucan glycosyltransferase in the “Glycosylphosphatidylinositol (GPI)-anchor biosynthesis” pathway. Similarly, the expression level of gene Ghir_D05G019060.1 was increased at least 30-fold in cotton induced by CEF08111.

## 3. Discussion

Recent years have witnessed the discovery of a new approaches to enhancement of the resistance of cotton to Verticillium wilt through cross protective. It is effective to use hypovirulent strains to induce and control Verticillium wilt in cotton. In this study, we confirmed that hypovirulent strain CEF08111 can significantly induce Verticillium wilt resistance in cotton, which is consistent with previous reports [18]. Based on comparative transcriptome analysis, we demonstrated that CEF08111 could induce Verticillium wilt resistance in cotton. At 3 hpi with CEF08111, the DEGs of susceptible cotton cultivar Jimian11 were mainly genes of the glycometabolism pathway compared with those of *V. dahliae* inoculated. Therefore, we speculate that saccharides secreted by CEF08111 were the key substances to induce resistance of cotton to *V. dahliae*. This speculation will be detected in subsequent trials. At 12 hpi, DEGs were mainly enriched in three signaling pathways: plant–pathogen interaction, MAPK signaling pathway—plant and plant hormone signal transduction. The pathways of plant–pathogen interaction and flavonoid biosynthesis were also induced in sunflower plants infected with *V. dahliae* [33], and the results were also consistent with those of Tan [34], who reported that most DEGs in tomato were associated with phenylpropanoid metabolism and plant–pathogen interaction pathways. However, the glutathione metabolism pathway has rarely been reported in the transcriptome of cotton plants treated with *V. dahliae*.

Plants have a series of defense mechanisms to respond to pathogen attack. PRRs are the first line of defense [35,36]; these receptors recognize pathogens and activate a resistance response [37]. In our study, 12 *EIX1/2* genes were significantly down-regulated at 12 hpi (Appendix A). This result was inconsistent with the study by Zhang et al. [38]. We will investigate whether decreased expression of *EIX1/2* is indirectly related to the activation of xylan signaling and determine whether the decreased expression affects the resistance of cotton to Verticillium wilt.

After recognizing the infection of *V. dahliae*, cotton instantly activated a complex series of defense-associated signaling pathways. Ca^2+^ influx is considered to play significant role in the early downstream response of numerous PAMP sensing processes, resulting in local and systemic acquired resistance [39]. Ca^2+^ activates calcium-dependent protein kinases (*CDPKs*), which play important roles in plant responses to both abiotic stress and pathogens [40,41]. In this study, *CDPK* (Ghir_D04G009070) were expressed at high levels in cotton during the early stage of inoculation. This result is consistent with that of Zhang et al. [38]. In addition, FLS2 recognizes flg22 and subsequently activates downstream signaling pathways that involve WRKY TFs to promote defense responses against bacterial and fungal pathogens and nematodes [42,43]. In this research, 6 *WRKY* genes were specifically upregulated at 12 hpi, as shown by the hierarchical clustering of DEGs (Appendix A). Among them, the genes with IDs Ghir_A04G009050, Ghir_D04G013210 and Ghir_D11G011570 were upregulated more than two-fold in the CEF08111 treatment group compared with sterile water treatment group. These results suggest that these *WRKY* genes may activate a series of downstream PR genes and thus play pivotal roles in the resistance response of CEF08111 to cotton. Overall, our results suggest that PRRs activate and promote the expression of downstream *CDPKs* and *WRKY* TFs; induce the accumulation of reactive oxygen species; and cause the deposition of cystatin in the cell wall, thereby inducing PTI in cotton after inoculation CEF08111.

During long-term evolutionary interactions with plants, several pathogens successfully cause ETS by producing a number of effectors. Simultaneously, plants have evolved R genes that recognize these effectors and function through highly specific interactions between effectors and their corresponding NB-LRR class receptors [25,26]. The rice CCNB-LRR protein Pi-to can directly interact with Avr factors, which the LRR domain is able to directly recognize the effector Avrpita of *Magnaporthe oryzae* and induce ETI [44]. It has also been demonstrated that the NBS-LRR protein from *Arabidopsis thaliana* RPM1 confers resistance to *Pseudomonas syringae*. RPM1 is also involved in the onset of hypersensitive response (HR) [45]. Consistent with previous studies, our results showed that genes encoding RPM1 (Ghir_D05G026550, BGI_novel_G003083 and Ghir_D11G031570) were significantly upregulated in cotton at 12 hpi induced by CEF08111 (Appendix A). These genes may play a key role in the induction of resistance to *V. dahliae* infection by CEF08111 in cotton.

Phytohormones are known to be important in the regulation of defense responses in plants [46,47]. SA, a crucial regulator of plant–pathogen interactions, induces HR and SAR [48]. In this study, 154 DEGs were identified as being associated with Phytohormones (Appendix A). Interestingly, four *MYC2* genes (Ghir_A05G028310, Ghir_D03G018560, Ghir_D03G018560 and Ghir_D11G006950.1) involved in the JA signaling pathway were significantly upregulated in cotton after CEF08111 inoculation (Appendix A). Importantly, the expression of *MYC2* (Ghir_A05G028310) was significantly higher in cotton at 12 hpi, suggesting that *MYC2* may have a significant function in the response of cotton to CEF08111 inoculation. This result is consistent with the studies of Han et al. [49].

## 4. Materials and Methods

### 4.1. Fungal Strain Culture

Strains *G. nigrescens* CEF08111 and *V. dalhiae* Vd076 were cultured on potato dextrose agar (PDA) plates for 7 d, inoculated into liquid Czapek–Dox medium [50], and cultured in the dark at 25 °C and 150 rpm for 7 d. The mycelia were filtered out and removed, and the filtrate was subsequently diluted to a 1 × 10^7^ spores/mL spore suspension.

### 4.2. Cotton Inoculation Treatment

In this study, we used one kind of cotton cultivar (Jimian 11) as a test plant. The variety was susceptible to *V. dahliae*. The cultivation and inoculation treatment solution were prepared according to the methods of Zhu et al. [18], with some modifications. The seeds were sterilized with 70% alcohol for 1 min and then with 1.0% sodium hypochlorite for 5 min, after which the seeds were washed with sterile water 3 times. The cotton seeds planted in paper pots (6 cm in diameter and 10 cm in height, made up of newspaper and without bottom) filled with autoclaved substrate (*vol/vol*, vermiculite:sand = 6:4). The paper pots were placed on plastic trays (18 × 25 cm). The experiment was conducted in a greenhouse at 23–30 °C and 12-h photoperiod. Seedlings were inoculated with CEF08111 and Vd076 spore suspension (1 × 10^7^ spores/mL) 20 days after sowing, respectively. The seedings were inoculated by placing the paper pots onto a plate (7 cm in diameter) containing 10 mL of spores suspension and incubating for 40 min; the pots were then returned to the plastic trays. Seedlings dipped in sterile water were used as the control. Each treatment with three replications (*n* = 3) had 12 pots, and each pot contained five plants. Leaf samples were then collected at 3 h, 6 h, 12 h, 24 h, 48 h, 72 h and 96 h, and 5 leaves were also randomly collected at each time point for each biological replicate under each treatment.

### 4.3. Control Effect of the CEF08111 on Verticillium wilt of Cotton

The above mentioned seedlings inoculated with CEF08111 spores suspension inoculated Vd076 spores suspension (1 × 10^7^ spores/mL) at 96 h post inoculation, and investigated at 20 days post inoculation (dpi) with Vd076. The disease severity was rated according to a disease index that was based on a five-scale categorization of Verticillium wilt disease of cotton seedlings [18].

### 4.4. H_2_O_2_ Measurement

H_2_O_2_ content was determined using the method described by Deniz et al. [51]. Fresh cotton leaves (0.1 g) were homogenized in 1 mL of cold acetone. Then, H_2_O_2_ content was determined using a hydrogen peroxide assay kit (Solarbio, Beijing, China) and its absorbance was measured at 415 nm. Data are represented as the amount of H_2_O_2_ per gram leaf (μmol/g). All analyses had three biological replicates.

### 4.5. JA and SA Measurement

The JA and SA contents in cotton plant leaves were analyzed using the Plant JA ELISA Kit and the Plant SA ELISA Kit (Sino Best Biological Technology Co. Ltd., Wuhan, China).

### 4.6. RNA Sequencing (RNA-Seq)

An RNAprepPurePlant Plus kit (Tian Gen, Beijing, China) was used to extract RNA from cotton leaves. Electrophoresis was performed, and a NANODROP 2000 spectrophotometer was used to detect the concentration and quality of RNA. Transcriptome sequencing was performed for the 3 h, 12 h and 48 h samples. Gn3, Gn12 and Gn48 represented the 3 h, 12 h and 48 h samples in the CEF08111 treatment group, respectively, and Vd3, Vd12 and Vd48 represented the 3 h, 12 h and 48 h samples in the Vd076 treatment group, respectively, and SW3, SW12 and SW48 represented the 3 h, 12 h and 48 h samples in the sterile water treatment group, respectively. Three biological replicates were performed, and there were 27 samples. The construction of the DNA library and sequencing were performed by Beijing Genomics Institute (BGI). Data filtering was performed using SOAPnuke software(version 1.4.0) (BGI, Beijing, China). Clean reads were obtained by removing the reads containing adapters, reads with more than 5% N, and low-quality sequences. The clean reads were spliced and aligned to the reference *G.hirsutum* genome retrieved from the cotton genome website (http://cotton.hzau.edu.cn, accessed on 16 August 2021). The fragments per kilobase per transcript per million mapped reads (FPKM) values were calculated and used to estimate the effects of sequencing depth and gene length on the mapped read counts.

### 4.7. Screening and Analysis of Differentially Expressed Genes (DEGs)

The DEseq2 [52] was used to analyze DEGs in cotton leaves treated or nontreated with CEF08111 under the criteria of a corrected *p* value < 0.001 and an absolute log2 ratio ≥ 1. GO (Gene Ontology) terms and KEGG (Kyoto Encylopedia of Genes and Genomes) pathways were enriched by DEGs if the *p* values were <0.001. Resistance genes among the DEGs were predicted by a BLAST search of the Plant Resistance Gene (PRG) Database (identity ≥ 40, E-value < 1 × 10^−5^) [53]. TFs encoded by the DEGs were predicted (E-value < 1 × 10^−5^) according to the Plant Transcription Factor Database [54].

### 4.8. Gene Co-Expression Network Analysis

Gene co-expression network analysis was performed using the WGCNA package V1.48 [55]. Gene dendrograms were constructed with colors based on the correlations between the expression levels of genes and used to build clustering trees and to divide modules. In addition, the correlation between modules and samples was analyzed using WGCNA.

### 4.9. Quantitative Reverse-Transcription-PCR (qRT-PCR) Analysis

A total of 12 genes were randomly selected for RTqPCR to verify whether the trends in their expression were consistent with the transcriptome sequencing results. Primers are given in Appendix A. Data were collected from three replicate experiments, and the samples used for RTqPCR were the same as those used for RNA-Seq. RNA was extracted from sample leaves and reverse transcribed into cDNA. RTqPCR was performed via a Roche LightCycler 480 Real-Time System (Roche, Rotkreuz, Switzerland), and each PCR mixture (20 μL) consisted of 10 μL 2 × PerfectStart Green qPCR SuperMix (Tiangen, Beijing, China), 1.0 μL of each primer, 1 μL of cDNA and 7.0 μL of sterile water. Each sample involved at least three technical repeats. The PCR cycle consisted of an initial denaturation step of 94 °C for 30 s, followed by 45 cycles of 94 °C for 5 s, 55 °C for 15 s and 72 °C for 10 s. The cotton ubiquitin gene was used as the internal reference, and relative gene expression was calculated using the 2^−ΔCT^ method.

## 5. Conclusions

Herein, we confirmed that *G. nigrescens* CEF08111 could improve the resistance of cotton (Jimian 11) to Verticillium wilt, and reduced H_2_O_2_ content and increased JA and SA content in cotton leaves. Comparative transcriptomics was applied to elucidate the molecular mechanism of the biological control. A total of 3841 DEGs related to disease resistance were detected. These genes related to ROS burst, Ca^2+^ influx, JA metabolism, glycolysis, gluconeogenesis, phenylpropane and lignin synthesis. Therefore, the disease-resistance genes should be studied in greater depth in the future to elucidate their role in the CEF08111-induced resistance response to *V. dahliae* infection in cotton.

## Figures and Tables

**Figure 1 ijms-24-01480-f001:**
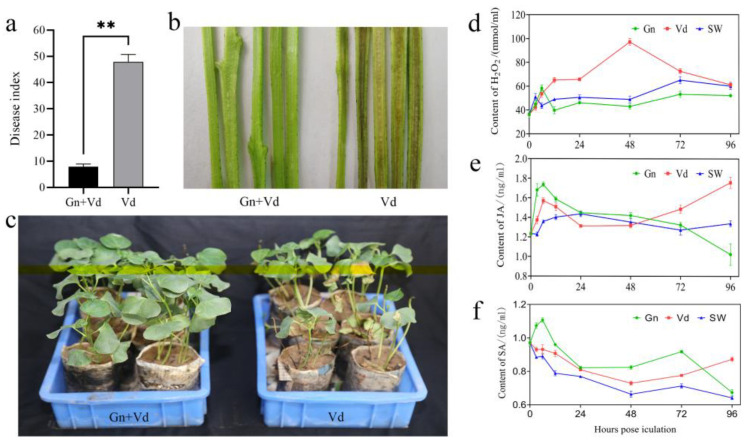
Disease index and symptoms of Verticillium wilt in cotton 20 d after *V. dahliae* inoculation. (**a**) The disease index (** *p* ≤ 0.01). (**b**) Vascular bundle discoloration. (**c**) Symptoms of Verticillium wilt in cotton. (**d**) The content of H_2_O_2_. (**e**) The content of JA. (**f**) The content of SA. Bars represent SEs.

**Figure 2 ijms-24-01480-f002:**
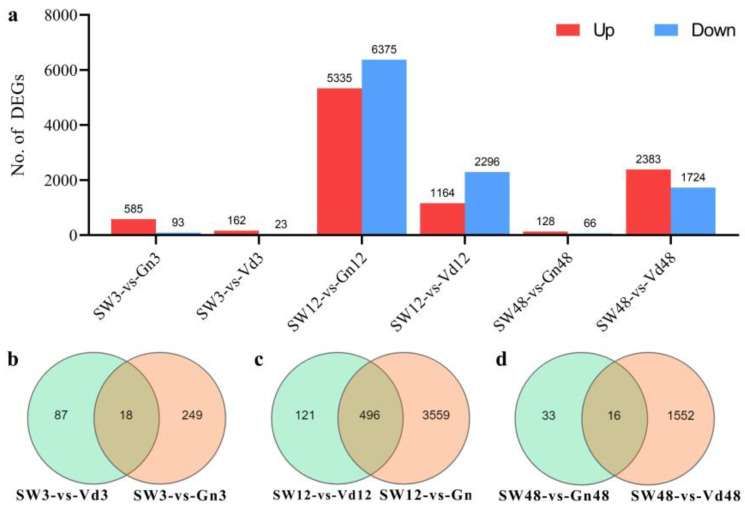
Total DEGs and their distribution in different comparisons post inoculation with CEF08111 and Vd076. (**a**) shows number of DEGs (up and down) in different inoculation treatments. (**b**–**d**) represent distribution of unique and common DEGs at 3 hpi, 12 hpi and 48 hpi, respectively.

**Figure 3 ijms-24-01480-f003:**
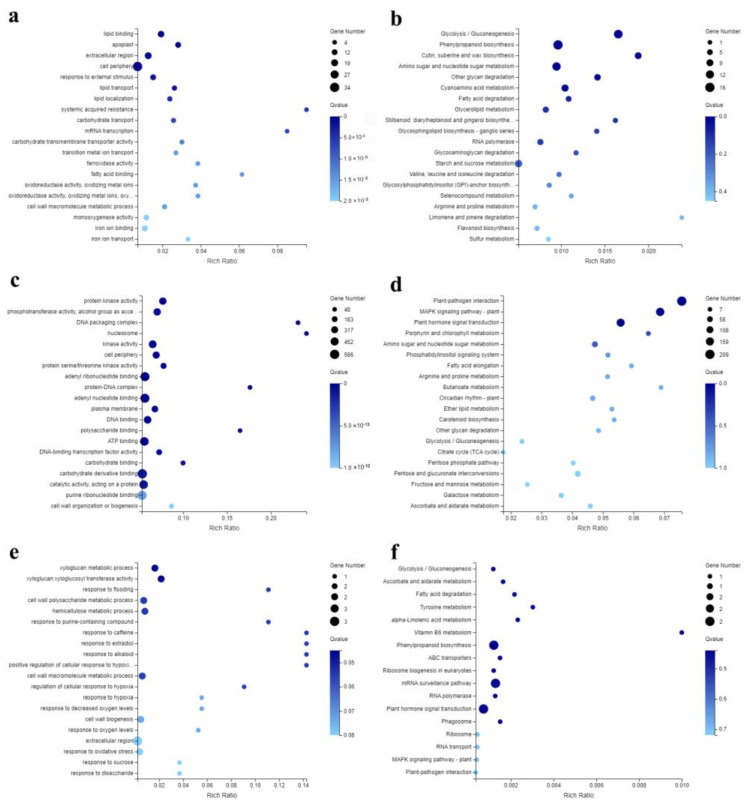
Scatter plot of GO and KEGG pathway enrichment of DEGs. The rich ratio is the ratio of the DEG number to the background number in a particular pathway. The size of the dots represents the number of genes, and the color of the dots represents the range of the Q-value. (**a**) GO pathways based on DEGs at 3 hpi. (**b**) KEGG pathways based on DEGs at 3 hpi. (**c**) GO pathways based on DEGs at 12 hpi. (**d**) KEGG pathways based on DEGs at 12 hpi. (**e**) GO pathways based on DEGs at 48 hpi. (**f**) KEGG pathways based on DEGs at 48 hpi.

**Figure 4 ijms-24-01480-f004:**
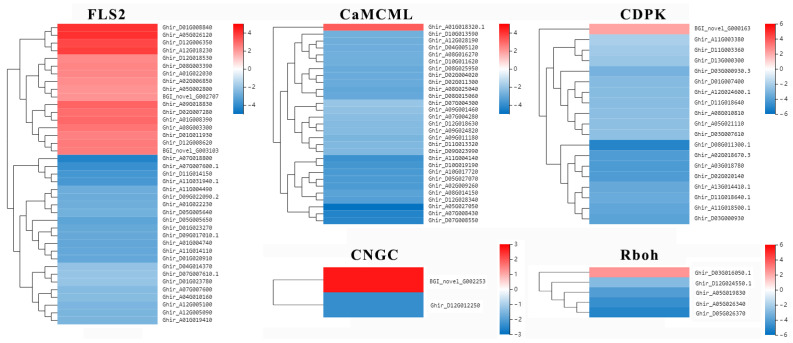
Expression levels of genes related to ROS and Ca^2+^. The red color represents upregulation, and the blue represents downregulation.

**Figure 5 ijms-24-01480-f005:**
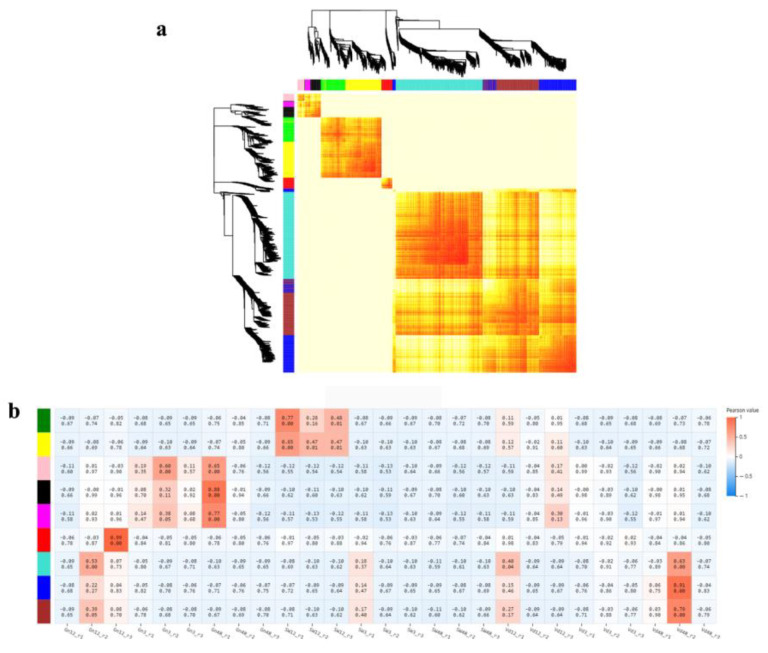
Gene co-expression network analysis by WGCNA. (**a**) Gene dendrogram colored according to correlations between gene expression levels. Different colors represent different gene modules and indicate coefficients of dissimilarity between genes. (**b**) Module–sample association. The abscissa represents the samples; the ordinate represents the modules. The numbers in each cell are the correlation coefficient (top) and *p*-value (bottom). The variation from blue (low) to orange (high) indicates the ranges of the DEGs.

**Figure 6 ijms-24-01480-f006:**
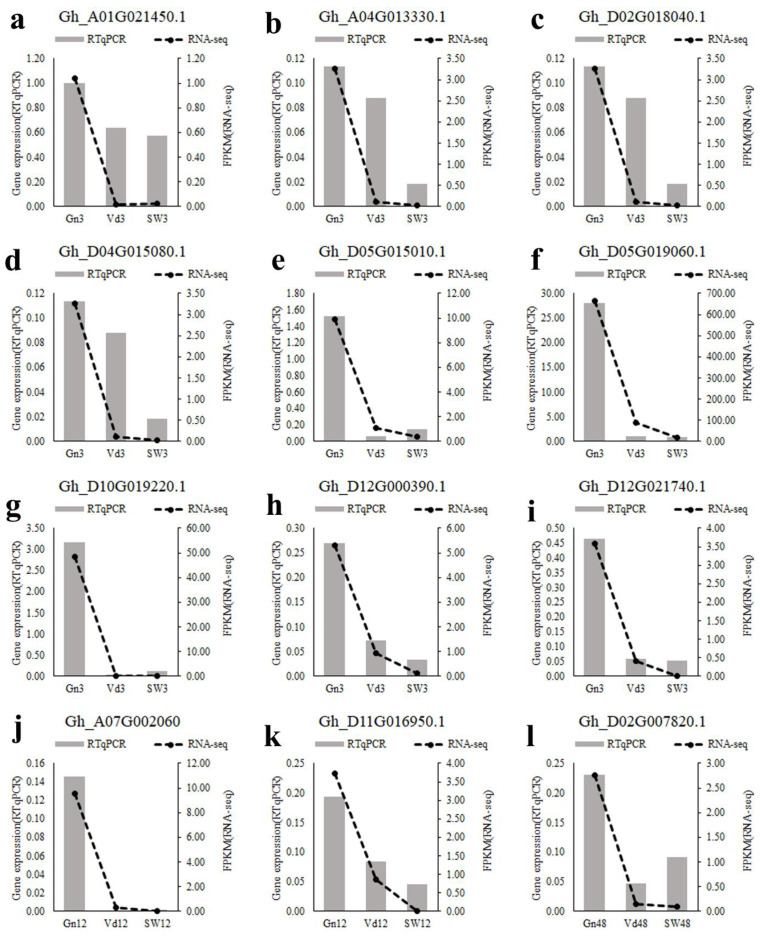
Comparison of the expression trends in the RTqPCR and RNA-Seq data. The gray bars represent gene expression levels relative to that of the cotton ubiquitin gene, which was used as an internal control, to normalize the expression levels of the target genes. Dotted lines represent the mean FPKM.

**Table 1 ijms-24-01480-t001:** KEGG pathway enrichment of 3841 DEGs.

Pathway ID	Pathway	Number of DEGs	*p*-Value	Q-Value
ko00010	Glycolysis/gluconeogenesis	14	9.84 × 10^−8^	6.99 × 10^−6^
ko00940	Phenylpropanoid biosynthesis	16	1.35 × 10^−5^	0.000320
ko00073	Cutin, suberine and wax biosynthesis	8	2.54 × 10^−5^	0.000450
ko00520	Amino sugar and nucleotide sugar metabolism	13	0.000111	0.001309
ko00511	Other glycan degradation	7	0.000465	0.004716
ko00460	Cyanoamino acid metabolism	9	0.000668	0.005926
ko04626	Plant–pathogen interaction	209	1.73 × 10^−24^	2.20 × 10^−22^
ko04016	MAPK signaling pathway—plant	166	1.68 × 10^−15^	1.06 × 10^−13^
ko04075	Plant hormone signal transduction	154	6.81 × 10^−8^	2.88 × 10^−6^

Pathways with a Q-value < 0.01 are shown.

## Data Availability

All data included in this study are available from the corresponding author.

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
