# Peer review of "Transcriptome Analysis Reveals the Defense Mechanism of Cotton against Verticillium dahliae Induced by Hypovirulent Fungus Gibellulopsis nigrescens CEF08111"

_ijms, 2023, doi:10.3390/ijms24021480_

Round 1

Reviewer 1 Report

Comment 1: In the title, write the scientific name in Italics.

Comment 2: Lot of abbreviations in the abstract, please rewrite with expansion clearly.

Comment 3: There is no information about Gibellulopasis nigrescen and related recent research in the introduction section, please include more information.

Comment 4: Lot of information in the results section but the abstract, Introduction, and Conclusion are not interesting for first-time readers. Please rewrite the above-said sections.

Comment 4: Is it Results or Results and Discussion?

Author Response

  1. In the title, write the scientific name in Italics.

Response: The scientific name Verticillium dahliae in the title is italicized.

  1. Lot of abbreviations in the abstract, please rewrite with expansion clearly.

Response: All abbreviations in the abstract were rewrite with expansion clearly.

  1. There is no information about Gibellulopasis nigrescens and related recent research in the introduction section, please include more information.

Response: We add the information about Gibellulopasis nigrescens and related recent research in the introduction section.

  1. Lot of information in the results section but the abstract, Introduction, and Conclusion are not interesting for first-time readers. Please rewrite the above-said sections.

Response: We have modified the content of abstract, Introduction, and Conclusion, and added some information.

  1. Is it Results or Results and Discussion?.

Response: We have revised the content of these two parts.

Reviewer 2 Report

As general comment the work is well written and designed with relevant results.

The authors touch upon very important issues about the defense mechanism of cotton against Verticillium dahliae induced by hypovirulent fungus Gibellulopsis nigrescens.

The progresses in our understanding of plant innate immunity make such a manuscript timely and I commend the authors for bringing in some new ideas and analysis.

This study is very interesting and conforms to the requirements of the International Journal of Molecular Sciences journal.

The issues discussed by the Authors are original.

The manuscript is very well written.

The format of the paper is correct.

The abstract and Introduction chapter are properly written.

Materials and method section is well described and correspond to the aim set out in the manuscript. 

The results are correctly described.

Figures and table are clear and understandable.

The discussion is correct.

The conclusions of the article are correct and consistent with the discussed issues.

The references are properly chosen and cited.

- I recommend the publication of this manuscript in the International Journal of Molecular Sciences journal in present form.

Author Response

Special thanks to you for your good comments.

Reviewer 3 Report

I checked your manuscript and described comments below.

Verticillium wilt is an important disease that damages crops.

I think this paper does a very good analysis of the resistance of cotton to Verticillium wilt by CEF08111.

I think that there is no problem with the content of this paper, but I think it would be better to correct the following points.

1.     There is no reference for WGCNA package 393 V1.48. "Langfelder P., and S. Horvath, 2008 WGCNA: An r package for weighted correlation network analysis. BMC Bioinformatics 9. https://doi.org/10.1186/1471-2105-9-559" as a reference is good.

2.     FS1.docx, TS4.xlsx, TS5.xlsx and TS7.xlsx among supplementary files are broken. I think it's better to change the file compression method to Zip or something like that.

I don't think this paper has any major mistakes or grammatical problems.

Author Response

  1. There is no reference for WGCNA package 393 V1.48. "Langfelder P., and S. Horvath, 2008 WGCNA: An r package for weighted correlation network analysis. BMC Bioinformatics 9. https://doi.org/10.1186/1471-2105-9-559" as a reference is good.

Response: We added the reference.

  1. FS1.docx, TS4.xlsx, TS5.xlsx and TS7.xlsx among supplementary files are broken. I think it's better to change the file compression method to Zip or something like that.

Response: We changed the file compression method to Zip.

Round 2

Reviewer 1 Report

Comment 1: There is no information about Gibellulopasis nigrescen and related recent research in the introduction section, please include more information.

Comment 2: ln84, when you write second time, include first letter of genus.

Comment 3: Improve discussion

Comment 4: Rewrite the conclusion

Author Response

We would like to thank you for your valuable comments again. All comments were carefully considered and the manuscript was revised accordingly, marked using MS Word's "Track changes" feature.

  1. There is no information about Gibellulopasis nigrescens and related recent research in the introduction section, please include more information.

Response: We add the information about Gibellulopasis nigrescens and related recent research in the introduction section.

  1. ln84, when you write second time, include first letter of genus.

Response: We have modified this problem in the paper and checked other similar problems in the whole paper.

  1. Improve discussion.

Response: We revised the whole discussion section.

  1. Rewrite the conclusion.

Response: We Rewrote the conclusion.
